# Biomarker Profiling of Upper Tract Urothelial Carcinoma Only and with Synchronous or Metachronous Bladder Cancer

**DOI:** 10.3390/biomedicines12092154

**Published:** 2024-09-23

**Authors:** Sara Meireles, Carolina Dias, Diana Martins, Ana Marques, Nuno Dias, Luís Pacheco-Figueiredo, João Silva, Carlos Martins Silva, Miguel Barbosa, Luís Costa, José Manuel Lopes, Paula Soares

**Affiliations:** 1Institute for Research and Innovation in Health (i3S), University of Porto, Rua Alfredo Allen 208, 4200-135 Porto, Portugal; anad@i3s.up.pt (C.D.); psoares@ipatimup.pt (P.S.); 2Institute of Molecular Pathology and Immunology of the University of Porto (IPATIMUP), Rua Júlio Amaral de Carvalho 45, 4200-135 Porto, Portugal; 3Faculty of Medicine, University of Porto, Alameda Professor Hernâni Monteiro, 4200-319 Porto, Portugal; 4Medical Oncology Department, Centro Hospitalar Universitário de São João (CHUSJ), Alameda Professor Hernâni Monteiro, 4200-319 Porto, Portugal; 5Pathology Department, Centro Hospitalar Universitário de São João, Alameda Professor Hernâni Monteiro, 4200-319 Porto, Portugal; 6Urology Department, Centro Hospitalar Universitário de São João (CHUSJ), Alameda Professor Hernâni Monteiro, 4200-319 Porto, Portugal; 7Life and Health Sciences Research Institute (ICVS), School of Medicine, University of Minho, 4710-057 Braga, Portugal; 8Department of Urology, Trofa Saúde Private Hospitals, 4785-409 Trofa, Portugal; 9Medical Oncology Department, Centro Hospitalar Universitário Lisboa Norte, Avenida Professor Egas Moniz MB, 1649-028 Lisboa, Portugal; 10Institute of Molecular Medicine—João Lobo Antunes, Faculty of Medicine, University of Lisbon, Avenida Professor Egas Moniz MB, 1649-028 Lisboa, Portugal

**Keywords:** upper tract urothelial carcinoma, molecular profiles, biomarkers, immunohistochemical stratification, bladder cancer

## Abstract

Background: Molecular profiling in upper tract urothelial carcinoma (UTUC) with synchronous or metachronous urothelial bladder cancer (UBC) is scarce. We intended to assess immunohistochemical (IHC) and genetic differences between UTUC-only and UTUC with synchronous or metachronous UBC (UTUC + UBC) and evaluate the effect of subsequent UBC on the outcome of UTUC patients stratified by luminal-basal subtypes. Methods: A retrospective cohort of UTUC was divided into UTUC-only (*n* = 71) and UTUC + UBC (*n* = 43). IHC expression of cytokeratin 5/6 (CK5/6), CK20, GATA3, and p53 was evaluated to assess relevant subtypes. Genetic characterization comprised *TERT*p, *FGFR3*, *RAS*, and *TP53* status. Kaplan–Meier and Cox regression analyses estimated the effect of clinicopathological variables and molecular profiles on progression-free survival (PFS) and overall survival (OS) of UTUC patients. Results: No meaningful differences were detected among both subgroups according to luminal-basal stratification and genetic analysis. UTUC + UBC was independently associated with a worse PFS when stratified by luminal-basal phenotype (HR 3.570, CI 95% 1.508–8.453, *p* = 0.004) but with no impact in OS (HR 1.279, CI 95% 0.513–3.190, *p* = 0.597). Conclusions: This study reveals that both subgroups exhibited equivalent genomic features and luminal-basal subtypes. The involvement of the bladder relates to shorter PFS but does not seem to influence the survival outcome of UTUC, independently of the IHC phenotype.

## 1. Introduction

Upper tract urothelial carcinoma (UTUC) represents a rare entity that arises in the urothelium of the renal pelvis or ureter, accounting for 5–10% of all urothelial carcinoma (UC) [1]. Given the more aggressive phenotype of UTUC compared with bladder cancer [2], a better understanding of their molecular landscape is mandatory to identify predictive biomarkers and potential therapeutic targets.

In urothelial bladder carcinoma (UBC), genomic analyses have identified distinct molecular subtypes, and a luminal-basal-like stratification was proposed [3,4,5,6,7,8]. Some authors report that primary UTUC and UBC display different phenotypes supporting an independent oncogenic pathway [9]. Further data showed that the two entities share similar genetic mutations but at varying frequencies [10,11,12,13,14]. Additionally, metachronous bladder tumors in paired UTUC patients appear to uphold the molecular features of the initial UTUC, which seems to favor the intraluminal seeding theory for developing bladder disease in these patients [15].

There is no substantial biomarker profiling data in primary UTUC with synchronous or metachronous UBC. Indeed, a stratification system in UTUC patients incorporating immunohistochemical (IHC) phenotype and gene expression analysis might be more reasonable to overcome the gap between molecular-based classification and the classic pathologic/immunohistochemistry-based classification [16]. However, the application of immunohistochemistry for a basic assessment of luminal-basal stratification in UTUC patients is pending for use in clinical practice.

This study intends to identify biological differences among patients with UTUC-only and UTUC with synchronous or metachronous bladder cancer (UTUC-UBC) based on IHC and gene expression analysis. We also aim to understand the putative effect of synchronous or metachronous bladder cancer on the outcome of UTUC patients after adjusting for luminal-basal phenotype with an immunohistochemistry-based protocol.

## 2. Materials and Methods

### 2.1. Study Design

The authors assembled a dataset of tissue samples from a retrospective cohort of patients diagnosed with UTUC in our institution between January 2009 and December 2019. Patients with synchronous (within six months) or metachronous UBC diagnosis were included in this study. The cases were divided into two subgroups: (1) UTUC-only (*n* = 71); (2) UTUC with synchronous or metachronous UBC diagnosis (UTUC + UBC; *n* = 43). Previous UBC diagnoses and patients submitted to neoadjuvant or adjuvant systemic treatment were excluded criteria.

Baseline demographic, clinicopathological, and outcome data were collected in a database through a comprehensive review of the patient’s electronic medical records.

The preoperative staging comprised urethrocystoscopy, which was used to rule out concomitant UBC, computed tomography (CT)/magnetic resonance imaging (MRI), and flexible ureteroscopy with biopsy. Postoperative surveillance was conducted according to European Association of Urology recommendations: cystoscopy and urine cytology every three months for the first two years, every six months for three years, and annually after that; abdominal and chest CT or MRI was considered within six months after surgery and then at least annually for a minimum of five years, depending on the clinical stage.

The following procedures were performed with the approval of the ethics regulatory hospital commission, granted on 3 February 2020 (process n° 425/19), and follow the recommendations of the Helsinki and Tokyo Declarations, the World Health Organization (WHO), and the European Community regulation.

### 2.2. Tumor Specimens

Representative hematoxylin and eosin (H&E) stained sections from all archived formalin-fixed paraffin-embedded (FFPE) tumor samples from UTUC patients submitted to radical nephroureterectomy resection or kidney-sparing approach were reviewed by an expert genitourinary pathologist. All the tissue samples containing adequate amounts of the tumor were staged following the 2002 American Joint Committee on Cancer (AJCC) staging system and graded based on the WHO pathological grading system of malignant urothelial cancer in 2004.

### 2.3. DNA Extraction

After histopathological examination, genomic DNA was extracted according to the manufacturer’s protocol (GRS Genomic DNA Kit, GRiSP, Porto, Portugal). Manual microdissection of the FFPE tissue block was performed using H&E slides as a template. The isolated DNA was stored at −20 °C or 4 °C for immediate use.

### 2.4. Immunohistochemical Analysis

The protein expression of markers cytokeratin 5/6 (CK5/6), CK20, GATA3, and p53 was evaluated to assess clinically relevant subtypes. Immunohistochemistry (IHQ) was performed using UltraVision™ Quanto Detection System HRP (REF: TL-125-QHL, Thermo Scientific, Waltham, MA, USA). Tissue sections were deparaffinized, rehydrated, and subjected to a 30 min treatment at 90 °C in 10× concentrated Epitope Retrieval Solution pH 9.0 (Ref. RE7119, Novocastra^TM^, Leica Biosystems, Newcastle Upon Tyne, UK). For p53 IHQ, antigen retrieval was performed in 10 mM sodium citrate buffer at pH 6.0. Endogenous peroxidase activity was inhibited with Ultravision™ hydrogen peroxide block (REF. TA-125-H202Q, Thermo Scientific, Waltham, MA, USA) and UltraVision™ protein block (REF. TA-125-PBQ, Thermo Scientific, Waltham, MA, USA). Then, slides were incubated overnight in a humid chamber at 4 °C with the monoclonal antibodies against CK5/6 (dilution 1:100, clone D5/16 B4, Dako, Agilent, Santa Clara, CA, USA) and CK20 (dilution 1:100, clone Ks20.8, Dako, Agilent, Santa Clara, CA, USA). Sections with anti-p53 antibody (dilution 1:600, clone DO-7, Leica Biosystems, Newcastle Upon Tyne, UK) were incubated at room temperature for 60 min. After rinsing with PBS, slides were incubated with HRP Polymer Quanto (REF. TL-125-QPH, Thermo Scientific, Waltham, MA, USA) followed by 3% diaminobenzidine chromogen (DAB, REF. TA-004-QHCX, Thermo Scientific, Waltham, MA, USA) and DAB Quanto Substrate (REF. TA-125-QHSX, Thermo Scientific, Massachusetts, USA) for chromogenic visualization. Regarding CK5/6, slides were incubated with HIGHDEF^®^ Red IHC Chromogen (HRP, REF. ADI-950-210-0030, Enzo Life Sciences, Farmingdale, NY, USA). Finally, slides were counterstained with Gill’s hematoxylin (Thermo Scientific, Waltham, MA, USA), cleared, and mounted.

Immunostaining for GATA3 (ready-to-use, clone L50-823, Master-inVitro diagnóstica) was conducted using a Ventana Benchmark XT automated staining system (Ventana Medical Systems, Tucson, AZ, USA). Slides were developed using the OptiView DABv3 detection kit (Roche, Rotkreuz, Switzerland), per the manufacturer’s instructions.

The expression for CK5/6, CK20, and GATA3 was evaluated semi-quantitatively according to the staining intensity (absent = 0, faint = 1, moderate = 2, or strong = 3) and proportion of positive-stained tumor cells (scored as <5% = 0; 5–25% = 1; 25–50% = 2; 50–75% = 3; and >75% = 4) [17]. The immunoreactive score (IRS) was defined by multiplying the intensity and proportion of positively stained cells. The positivity cut-off was established based on the median of IRS values (CK5/6 > 3, CK20 > 6, GATA3 > 8). Cases stained for p53 were classified as follows: wild-type (1–49% nuclear expression) or aberrant (null-phenotype: 0% nuclear expression; 50–99% nuclear expression; or diffuse overexpression: 100% nuclear expression) [18].

Luminal-like subtype was defined as CK20+ or GATA3+/CK5/6− and basal-like as CK20− or GATA3−/CK5/6+.

### 2.5. Targeted Sequencing Genomic Characterization

The genetic characterization of UTUC was performed by analyzing mutations frequently reported in UC, specifically in *TERT*p, *FGFR3*, *RAS* (*HRAS*, *KRAS*, and *NRAS*), and *TP53*.

Detection of target hotspot mutations in the promotor region of the *TERT* gene (NM_198253; in the −124 and −146 positions to the transcription start site) and in the *FGFR3* gene (NM_000142; in exon seven at codons 248 and 249) was performed by quantitative real-time polymerase chain reaction (PCR) (QuantStudioTM 5 Real-Time PCR System, Applied Biosystems, Waltham, MA, USA), using primers and probes provided by the Uromonitor^®^ test kit (U-monitor, Porto, Portugal) and according to the manufacturer’s protocol.

Frequently mutated regions on the exons 5–9 of the *TP53* gene (NM_000546) and in codons 12, 13, and 61 of the *HRAS* (NM_005343) and the *KRAS* (NM_004985) genes were analyzed by Sanger sequencing. For the *NRAS* gene (NM_002524), only codon 61 was analyzed.

Genomic DNA (25–50 ng) was amplified using the QIAGEN multiplex PCR kit (QIAGEN, Hilden, Germany) for *TP53* and the MyTaq HS Mix 2X Bioline PCR kit (Meridian Bioscience, Cincinnati, OH, USA) for *RAS*, following the manufacturer’s instructions. The annealing temperature of 60 °C was established for *NRAS* segment amplification, while *TP53*, *HRAS*, and *KRAS* were screened separately by a touchdown PCR. After confirming DNA amplification with a 1% agarose gel electrophoresis (GRS Agarose LE, GRiSP, Porto, Portugal), PCR products were purified with Exonuclease I (Thermo Scientific, Waltham, MA, USA) and Shrimp Alkaline Phosphatase (Thermo Scientific, Massachusetts, USA) and sequenced using the Big Dye Terminator v3.1 Cycle Sequencing Kit (Applied Biosystems, Waltham, MA, USA). After precipitation, sequencing products were separated by capillary electrophoresis and analyzed in an automatic sequencer (ABI PRISM 3100 Genetic Analyzer, Perkin-Elmer, Foster City, CA, USA). All identified mutations were confirmed and validated by an independent PCR analysis.

### 2.6. Outcomes and Statistical Analysis

All statistical analyses were performed using the Statistical Package for Social Sciences (SPSS, IBM Corp., Chicago, IL, USA) software, version 28.0.

Disease progression was defined as a newly diagnosed local disease, bladder recurrence, or distant metastasis. Overall survival (OS) and progression-free survival (PFS) were stated as the time between the date of diagnosis and the date of death/disease progression or the end of follow-up.

Univariate analysis was performed using the appropriate Chi-square or Fisher’s exact test. Student’s unpaired *t*-test was also conducted for independent samples or the Wilcoxon signed-rank test if the data were not normally distributed.

Overall survival and PFS were summarized using the Kaplan–Meier method, and the log-rank test was used to assess differences between subgroups. Multivariate analysis with the Cox regression model assessed the effect of different clinicopathological variables and molecular profiles on the survival analysis. Differences were considered statistically significant when *p*-value < 0.05.

Missing data can be due to a lack of available information or data technically infeasible.

## 3. Results

### 3.1. Clinicopathological Characteristics of UTUC Patients

One hundred fourteen patients with UTUC diagnoses were included in the present study. The median follow-up time was 23 months (IQR: 10–50). A total of 35 patients (35.4%) experienced disease progression, and 36.8% (*n* = 42) died of the disease. Twenty-five patients (21.9%) were lost to follow-up.

The median age at diagnosis was 75 years (interquartile range, IQR: 66–80), and 69.3% (n = 79) were male. Most patients had stage ≥ II (69.9%, *n* = 79) and a high-grade tumor (93.8%, *n* = 105). A total of 43 patients had a history of synchronous (12.3%, *n* = 14) or metachronous UBC (25.4%, *n* = 29), and among them, 9% (*n* = 4) were diagnosed in the muscle-invasive stage. Demographic and clinicopathological data of UTUC patients are summarized in Table 1.

Both subgroups, UTUC-only and UTUC+UBC, were comparable regarding the defined clinicopathologic variables (Table 1). Still, there were no statistically significant differences among higher-risk features such as lymphovascular invasion (LVI) (*p* = 0.127), non-pure urothelial variant (*p* = 0.502), and lymph node involvement (*p* = 0.099). Metastasis at diagnosis was significantly higher in the UTUC-only subgroup than in UTUC + UBC (16.9 vs. 2.3%, *p* = 0.03). These patients (*n* = 13) were excluded from the survival analysis.

### 3.2. Immunohistochemical and Gene Expression Profiling and Their Association with Clinicopathological Features

Our series was described regarding the expression of IHC markers and the selected gene’s status. Overall, positive expression of CK5/6, CK20, and GATA3 was identified in 47.4% (*n* = 54), 49.1% (*n* = 56), and 46.5% (*n* = 53) of UTUC cases, respectively (Figure 1). Aberrant expression of p53 was found in 19.5% (*n* = 22) cases.

Across all UTUC samples, 76.3% (*n* = 87) were mutated for at least one of the studied genes. Specifically, the frequency of *FGFR3*, *TERTp*, and *RAS* mutations was 48.7% (*n* = 55), 52.7% (*n* = 58), and 7.9% (*n* = 9), respectively. Co-occurring genomic alterations were identified among *FGFR3*-*TERTp* (32.7%, *n* = 36), *FGFR3*-*RAS* (3.6%, *n* = 4), and *RAS-TERTp* genes (3.6%, *n* = 4). An overview of the genomic landscape and mutational profile of UTUC samples is shown in Figure 2.

To further elucidate the aberrant p53 IHC pattern, we evaluated *TP53* mutations in a subset of cases with aberrant p53 expression (*n* = 20) and in a similar number of cases with wild-type expression status (*n* = 20). *TP53* mutation was found in 12 (30%) cases, 5 with wild-type expression pattern and 7 with aberrant type expression pattern. Thus, 35% of patients with *TP53* mutation and 65% with wild-type status were associated with an aberrant p53 IHC pattern (Appendix A).

The relationship between molecular profiling and clinicopathological variables was also assessed (Appendix A). A higher tumor stage was associated with CK20 positive expression (*p* = 0.025) and an aberrant expression pattern for p53 (*p* = 0.003). Instead, LVI (*p* = 0.006), lymph node involvement (*p* = 0.047), and metastasis at diagnosis (*p* = 0.046) were detected in tumors with negative CK20 expression. A pure urothelial histological subtype (*p* = 0.029) was related to GATA3 positive expression (*p* = 0.029) and CK5/6 expression (*p* = 0.046). Inversely, *RAS* mutations were frequently found in non-pure UC (*p* = 0.001). *FGFR3* mutation was commonly linked with the absence of LVI and carcinoma in situ and *TERTp* mutation with larger tumor size (*p* = 0.048).

A total of 50 cases were assigned to the luminal-like subtype (43.9%) and 48 patients to the basal-like subtype (42.1%). There was no association between *FGFR3*, *TERTp*, and *RAS* expression and luminal-basal stratification.

### 3.3. Molecular Profiling Comparison between UTUC-Only and UTUC + UBC Subgroups

Immunohistochemical and genomic expression were compared to evaluate the molecular differences between UTUC-only and UTUC + UBC subgroups. No meaningful differences were detected among luminal and basal-like stratification and gene expression analysis between UTUC-only and UTUC + UBC (all *p* values > 0.05). Detailed IHC and gene expression profiling in both subgroups are summarized in Table 2.

### 3.4. Survival Analysis of UTUC-Only and UTUC + UBC Subgroups According to Luminal-Basal Phenotype

Overall, the luminal-like subtype showed a better PFS than the basal-like subtype (2y-PFS: 79.9% vs. 58.7%, *p* = 0.038) in the entire UTUC series, but OS was not statistically different (2y-OS: 64.5% vs. 62.9%, *p* = 0.499) (Appendix A).

*FGFR3*, *TERTp,* and *RAS* status were not associated with significant differences in PFS and OS of UTUC patients (Appendix A).

UTUC-only had consistently higher PFS than UTUC + UBC subgroup, either in the luminal-like subtype (2y-PFS 83.5% vs. 41%, *p* = 0.014) or basal-like phenotype (2y-PFS: 68.1% vs. 45.2%, *p* = 0.028) (Figure 3). Overall survival was not distinct for both UTUC subgroups when stratified by luminal-basal phenotypes (luminal: UTUC-only 2y-OS 68.8% vs. UTUC + UBC 74.8%, *p* = 0.288; basal: UTUC-only 2y-OS 70.1% vs. UTUC + UBC 75.4%, *p* = 0.584) (Figure 4).

### 3.5. The Impact of Bladder Cancer in UTUC Outcome Stratified by Luminal-Basal Phenotype

Synchronous or metachronous UBC was independently associated with a worse PFS when stratified by luminal-basal phenotype (HR 3.570, CI 95% 1.508–8.453, *p* = 0.004) (Table 3). However, it had no impact on OS, adjusting for IHC phenotype (HR 1.279, CI 95% 0.513–3.190, *p* = 0.597) (Table 3). The histological subtype (*p* = 0.003), LVI (*p* < 0.001), AJCC staging system (*p* = 0.031), and p53 status (*p* = 0.034) were identified as significant prognostic factors for PFS on univariate analysis (Table 3). Nevertheless, only LVI was independently associated with decreased PFS (HR 3.265, CI 95% 1.378–7.734 *p* = 0.007) according to luminal-basal stratification (Table 3).

Smoking status (*p* = 0.029), LVI (*p* < 0.001), lymph node involvement (*p* = 0.035), aberrant p53 (*p* = 0.048), and non-pure histological variant (*p* = 0.006) showed a negative association for OS in univariate analysis (Table 4). Multivariate analysis, adjusted for luminal-basal phenotypes, disclosed smoking status (HR 4.256, CI 95% 1.303–13.898, *p* = 0.016) and LVI (HR 13.510, CI 95% 3.837–47.570, *p* < 0.001) as prognostic factors independently related to worse OS (Table 4).

## 4. Discussion

Molecular evaluation in UC reveals potential prognostic and predictive biomarkers and can assign a better patient selection for tailored treatment strategies. A comprehensive classification including IHC and genetic expression might be more effective in subtyping UC.

In the present study, we conducted a systematic analysis comprising IHC and the genetic profile of a retrospective UTUC series to assess putative biological differences between two subgroups of UTUC: UTUC-only and UTUC with synchronous and metachronous UBC.

We reported previously that UTUC-only and UTUC with concomitant or later UBC share clinicopathologic features [19]. Here, we substantiate that the phenotypical and genomic (*FGFR3*, *RAS*, and *TERTp*) characteristics among both primary UTUC subgroups are comparable. Likewise, our study reveals that both subgroups exhibited equivalent luminal and basal-like subtypes and that the involvement of the bladder does not seem to influence the survival outcome of UTUC independently of the IHC phenotype.

Over the last few years, many studies have expanded our understanding of the genomic characterization of UTUC. However, a highly complex and costly advanced molecular technology limits a comprehensive molecular analysis’s applicability in clinical settings. Moreover, immunohistochemistry has the unique advantage of disclosing the precise morphological distribution of biomarkers in cell/tissue components [20]. A strong correlation between IHC features and mRNA expression profiles in basal (CK5/6-positive) and luminal (CK20-positive) subtypes of UC was recently reported [21,22]. The latest evidence has also reported that molecular subtypes in UBC may be recognized with GATA3 and CK5/6 expression in 80–90% of cases [23,24,25]. Nonetheless, an immunohistochemistry-based method for a critical assessment of luminal-basal stratification in UTUC still needs to be approved for routine clinical practice.

Firstly, we validated the luminal-basal-based method for risk stratification in our UTUC series with an IHC protocol, combining the expression of CK5/6, GATA3, and CK20 markers according to proposed patterns in UBC [16,26]. A comparable representation of luminal-like and basal-like subtypes was uncovered in the entire cohort. As expected, luminal-like phenotype bared a lower risk of disease progression, and important aggressive clinicopathologic features (LVI, lymph node involvement, and metastasis at diagnosis) were associated with negative CK20 status. Unpredictably, CK20 positivity was related to a higher tumor stage, and CK5/6 positive expression was linked to variant histology with favorable clinical outcomes. The same misleading prognostic results were already revealed in series with non-muscle invasive bladder cancer (NMIBC) [8]; furthermore, Sikic et al. [22]. reported that in non-muscle-invasive high-grade UTUC, both CK5-negativity and CK20-positivity were associated with poor prognosis. These results are in contradiction with prior findings in muscle-invasive bladder cancer (MIBC) and breast cancer, in which basal markers were linked with meaningful worse outcomes [27]. The debate remains on whether IHC-based patterns in non-muscle-invasive UTUC are more closely related to those of NMIBC.

Most of the available molecular data have reported slight genomic differences between primary UTUC and primary genetically unrelated UBC [11,14], indicating that UTUC seems to display more frequent mutations in *FGFR3* and *HRAS*, whereas UBC harbors *TP53*, *RB1*, *ATM*, and *ERBB2* mutations [11,12,13,14,28]. In our UTUC series, we have uncovered *FGFR3* mutations as one of the significant genetic events, as reported before in other analyses [11,29,30]. Notably, we identified a higher prevalence of mutations at the p.R248C hotspot than the common mutations at codon 249 [29,31,32]. We also found that *FGFR3* alterations were linked with the absence of LVI and carcinoma in situ, supporting their known effect as biomarkers associated with better outcomes [29,30,33].

The *TERTp* mutation frequency that we observed in the present study was described in UTUC and UBC [34]. Our results were aligned with the Killela et al. report on UTUC patients (52.7 vs. 47.3%) [35]. These somatic mutations have been related to higher tumor progression and distant metastasis in several cancers, but discordant findings exist regarding UC [34,36]. However, our study barely associated a higher tumor size with *TERTp* alterations.

*KRAS* mutations represented the most common mutated RAS gene, and few patients displayed *HRAS* mutations, at variance with findings in previous UTUC series (12%) [11,30,37,38]. Since our cohort included tumors with more aggressive features, a higher prevalence of *KRAS* mutations could be expected.

Although we have not conducted *TP53* mutational analysis in the entire tumor cohort, our findings support their low incidence in UTUC compared with UBC, as reported in other studies (30 vs. 57.8%) [11]. Also of note is the fact that p53 IHC is not a direct read-out of the gene status. We found cases with non-aberrant p53 IHC expression that harbor *TP53* mutation and *TP53* non-mutated cases showing aberrant IHC expression. It is essential to recognize that our evaluation has focused solely on the hotspot genetic region of the *TP53*, meaning that we cannot rule out the possibility of additional mutations occurring in other regions of the *TP53* gene. Otherwise, we must consider the high complexity of cellular process regulation as well as molecular and tumor microenvironment interactions [39]. Further proteome-wide comprehensive analysis of the p53 IHC pattern would be necessary to better identify and acknowledge any such *TP53* dysfunctions.

Remarkably, our research team has recently published a molecular analysis of 125 NMIBC, showing an equivalent frequency of *TERTp* and *FGFR3* mutations in UBC samples [40]. These results indicate a high genetic similarity between primary UTUC and an unrelated primary UBC, which differs from the current literature-reported data [37].

Together, we postulated that UTUC might share genotype and phenotype expression profiles with NMIBC, as argued before [8,41,42,43]. However, it is crucial to state that different stages of the disease, comprising non-muscle-invasive and muscle-invasive tumors, have been evaluated in a published series of UTUC and UBC studies, making comparison challenging. Additionally, a distinct pathway seems to be involved in the disease pathogenesis of UTUC and UBC [9].

The clonal relatedness of primary UTUC with an intravesical recurrence or a divergent primary UBC remains unanswered. The biological idiosyncrasies of primary UTUC with concomitant or later UBC are also unknown. It has been proposed that molecular events in UTUC and UBC can be shared or distinct. Yet, our findings suggest a similar expression profile when analyzing both primary UTUC subgroups.

In the era of molecular subtypes in UC, the impact of subsequent UBC on the outcome of primary UTUC, with no previous bladder cancer diagnosis, is also debatable. We sustain the concept that UTUC itself might display an aggressive phenotypical course and could be the main influential factor, independently of the presence of UBC. Despite the higher risk of recurrence, the effect of bladder cancer seems not to establish a worse prognosis, even when adjusted for luminal-basal stratification and genetic alterations.

Several studies showed that UTUC patients with concomitant or bladder cancer histories had a worse prognosis [44,45,46]. Still, the same assumption in patients with no previous bladder cancer diagnosis has not yet been systematically evaluated. On the other hand, a meticulous clinical follow-up of patients with previous tumor history could support a better prognosis. Divergent results regarding intravesical recurrence (IVR) and the impact on the prognosis of UTUC have also been reported [44,47,48,49,50]. Kuroiwa et al. refined that IVR was not associated with worse survival in UTUC patients with pT3–4 disease stage [44].

However, significant confounding factors of this study comprise a higher disease stage of our UTUC patients and the different proportion of patients submitted to nephroureterectomy in both subgroups, compared with other reports.

Noteworthy, our study highlights the prognostic implication of LVI in the risk stratification of UTUC, as we reported in a previous study [19]. These results also emphasize a strong negative impact of smoking in OS, while there was no impact on disease progression. Perhaps these findings underlie the adverse association of smoking with the poor global health status of the patients instead of representing a direct effect on disease course.

Our study had some inherent limitations. The cohort represents a single-center retrospective study with a relatively small sample of patients and short follow-up, which might influence the representativity and statistical power of the results. Moreover, our analysis was focused on DNA and protein-based analyses, whereas comprehensive large-scale DNA sequencing techniques may unravel other UTUC alterations. In addition, restrictions associated with immunohistochemistry, such as tumor heterogeneity and the extension and intensity of the marker’s expression, warrant further validation studies. Different scoring systems have been used beyond the IRS to quantify marker expression, leading to diverse results. Another reason for the other results might relate to a few specific markers used as surrogate markers for the UTUC subtypes. Yet, in our work, the pathologist’s intervention in selecting adequate samples for molecular analysis seems an improvement over other studies.

Despite our limitations, this study provides novel insights concerning UTUC molecular and IHC stratification that could be validated in more extensive prospective studies, guiding the selection of patients for a tailor-made approach.

Likewise, further research focused on UC’s tumor microenvironment will help us better understand tumor biology and how the immune landscape reflects on clinical outcomes and immunotherapy responses.

## 5. Conclusions

Reported data on molecular subtypes of UTUC remain an unmet need. Indeed, the assessment of biomarker profiling of UTUC with synchronous or metachronous bladder cancer is still limited.

Our study reveals similar luminal-basal phenotype and genetic features among patients with UTUC with synchronous or metachronous UBC and UTUC without UBC. An immunohistochemistry-based protocol was recognized as an accessible and standardized method for luminal-basal classification in UTUC patients.

Furthermore, in the era of molecular subtypes, the association of bladder cancer does not seem to convey a worse outcome for UTUC patients, even when adjusted for molecular and luminal-basal subtypes.

This study highlights the biological heterogeneity and molecular features of UTUC associated with an individual worse prognosis, independently of bladder involvement.

## Figures and Tables

**Figure 1 biomedicines-12-02154-f001:**
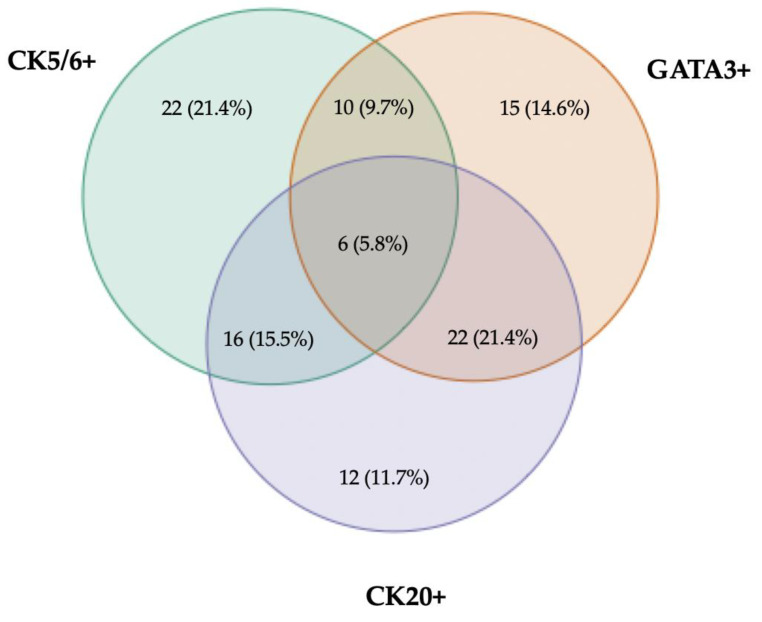
Venn diagram representing the positive expression of the CK5/6 (basal), GATA3, and CK20 (luminal) markers.

**Figure 2 biomedicines-12-02154-f002:**
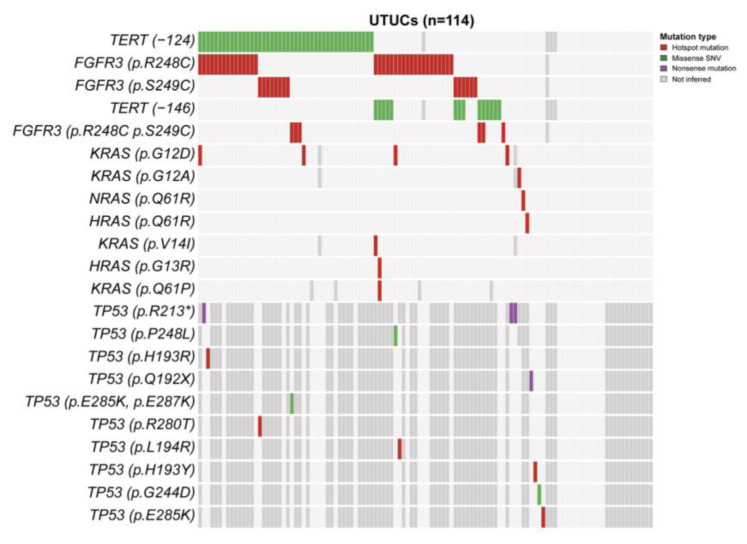
Heatmap depicting *TERT* promoter, *FGFR3*, *TP53*, *KRAS*, *HRAS*, and *NRAS* somatic mutations inferred in upper tract urothelial tumors through Sanger sequencing. Cases are shown in columns, and genes are in rows. The mutation types are color-coded according to the legend (right). Abbreviations: SNV—single-nucleotide variant; UTUCs—upper tract urothelial carcinomas. *TP53 n* = 40 cases.

**Figure 3 biomedicines-12-02154-f003:**
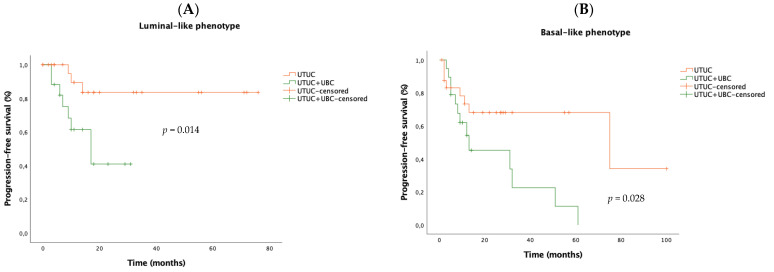
Kaplan–Meier curves comparing progression-free survival (PFS) between UTUC-only and UTUC + UBC patients in luminal (**A**) and basal (**B**) subtypes. Abbreviations: UBC—urothelial bladder cancer; UTUC—upper tract urothelial carcinoma; Log-rank test, statistical significance *p* value < 0.05.

**Figure 4 biomedicines-12-02154-f004:**
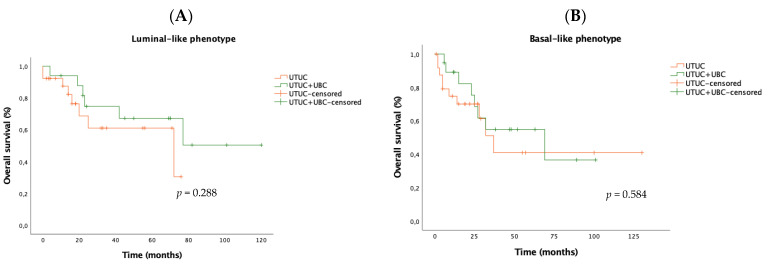
Kaplan–Meier curves comparing overall survival (OS) between UTUC-only and UTUC + UBC patients in luminal (**A**) and basal (**B**) subtypes. Abbreviations: UBC—urothelial bladder cancer; UTUC—upper tract urothelial carcinoma; Log-rank test, statistical significance *p* value < 0.05.

**Table 1 biomedicines-12-02154-t001:** Baseline clinicopathologic characteristics of upper tract urothelial carcinoma patients.

Baseline Characteristics	Total, n (%)	UTUC-Only, n (%)	UTUC + UBC, n (%)	*p*-Value
**Number of patients** (n/%)	114 (100)	71 (62.2)	43 (37.7)	
**Age** (median, years)	75 (41–94)	74 (41–91)	77 (55–94)	0.108 ^a^
**Gender**				0.933 ^b^
Male	79 (69.3)	49 (69)	30 (69.8)	
Female	35 (30.7)	22 (31)	13 (30.2)	
**Smoking**				0.941 ^b^
Yes	49 (55.1)	31 (55.4)	18 (54.5)	
No	40 (44.9)	25 (44.6)	15 (45.5)	
**Hydronephrosis**				0.901 ^b^
Yes	55 (49.5)	35 (50)	20 (48.8)	
No	56 (50.5)	35 (50)	21 (51.2)	
**Tumor location**				0.387 ^b^
Renal pelvis	65 (57)	44 (62.0)	21 (48.8)	
Ureter	33 (28.9)	18 (25.4)	15 (34.9)	
Both	16 (14.0)	9 (12.7)	7 (16.3)	
**Surgical procedure**				0.079 ^b^
Nephroureterectomy	105 (92.1)	68 (95.8)	37 (86.0)	
Kidney-sparing approach	9 (7.9)	3 (4.2)	6 (14)	
**Histological subtype**				0.502 ^c^
Pure UC	102 (89.5)	63 (88.7)	39 (90.7)	
Non-pure UC	12 (10.5)	8 (11.3)	4 (9.3)	
**Lymphadenectomy**				0.741 ^b^
Yes	30 (26.8)	18 (25.7)	12 (28.6)	
No	82 (73.2)	52 (74.3)	30 (71.4)	
**Tumor size, cm**				0.961 ^b^
≤2	16 (14.2)	10 (14.3)	6 (14)	
>2	97 (85.8)	60 (85.7)	37 (86)	
**Multifocality**				0.327 ^c^
Yes	103 (90.4)	5 (7)	6 (14)	
No	11 (9.6)	66 (93)	37 (86)	
**Tumor grade**				0.548 ^c^
Low-grade	7 (6.3)	4 (5.8)	3 (7.0)	
High-grade	105 (93.8)	65 (94.2)	40 (93)	
**Lymphovascular invasion**				0.127 ^b^
Yes	38 (34.2)	27 (39.7)	11 (25.6)	
No	73 (65.8)	41 (60.3)	32 (74.4)	
**Carcinoma in situ**				0.782 ^b^
Yes	20 (17.5)	13 (18.3)	7 (16.3)	
No	94 (82.5)	58 (81.7)	36 (83.7)	
**AJCC staging ***				0.878 ^b^
0is + 0a + I	34 (30.1)	21 (29.6)	13 (31)	
II + III + IV	79 (69.9)	50 (70.4)	29 (69)	
**Lymph node involvement**				0.099 ^b^
Yes	8 (26.7)	7 (38.9)	1 (8.3)	
No	22 (73.3)	11 (61.1)	11 (91.7)	
**Metastasis at diagnosis**				0.030 ^c^
Yes	13 (11.4)	12 (16.9)	1 (2.3)	
No	101 (88.6)	59 (83.1)	42 (97.7)	

% valid percent; * excluded patients with metastasis at diagnosis and locally advanced unresectable disease; ^a^ Student’s *t*-test; ^b^ Chi-square test; ^c^ Fisher’s exact test; Abbreviations: AJCC—American Joint Committee on Cancer; cm—centimeter; ECOG PS—Eastern Cooperative Oncology Group Performance Status; n—number of patients; UBC—urothelial bladder cancer; UC—urothelial carcinoma; UTUC—upper tract urothelial carcinoma.

**Table 2 biomedicines-12-02154-t002:** IHC staining and gene expression rates in UTUC subgroups (UTUC-only vs. UTUC + UBC).

IHC Markers	Total, n (%)	UTUC-Only, n (%)	UTUC + UBC, n (%)	*p*-Value
**CK5/6**				0.887 ^a^
Positive	54 (47.4)	34 (47.9)	20 (46.5)	
Negative	60 (52.6)	37 (52.1)	23 (53.5)	
**CK20**				0.059 ^a^
Positive	56 (49.1)	30 (42.3)	26 (60.5)	
Negative	58 (50.9)	41(57.7)	17 (39.5)	
**GATA3**				0.122 ^a^
Positive	53 (46.5)	37 (52.1)	16 (37.2)	
Negative	61 (53.5)	34 (47.9)	27 (62.8)	
**p53**				0.931 ^a^
Wild-type	91 (80.5)	57 (80.3)	34 (81)	
Aberrant	22 (19.5)	14 (19.7)	8 (19)	
**Luminal-basal phenotype**				0.715 ^a^
Luminal	50 (43.9)	32 (52.5)	18 (48.6)	
Basal	48 (42.1)	29 (47.5)	19 (51.4)	
**Gene Status**				
** *FGFR3* **				0.863 ^a^
Wild-type	58 (51.3)	36 (50.7)	22 (52.4)	
Mutated	55 (48.7)	35 (49.3)	20 (47.6)	
** *Specific mutations* **				
*Exon 7 p.R248C*	35 (31.0)	25 (35.2)	10 (23.8)	
*Exon 7 p.S249C*	14 (12.4)	7 (9.9)	7 (16.7)	
*Exon 7 p.R248C* + *p.S249C*	6 (5.3)	3 (4.2)	3 (7.1)	
** *TERTp* **				0.808 ^a^
Wild-type	52 (47.3)	33 (47.8)	19 (46.3)	
Mutated	58 (52.7)	36 (52.2)	22 (53.7)	
** *Specific mutations* **				
*c.1-124G*>*A*	44 (40)	28 (40.6)	16 (39.0)	
*c.1-146G*>*A*	14 (12.7)	8 (11.6)	6 (14.6)	
** *RAS* **				0.727 ^b^
Wild-type	105 (92.1)	66 (93)	39 (90.7)	
Mutated	9 (7.9)	5 (7)	4 (9.3)	
** *Specific mutations* **				
*NRAS p.Q61R*	1 (0.9)	1 (1.4)	0	
*HRAS p.G13R*	1 (0.9)	0	1 (2.3)	
*KRAS p.G12D*	4 (3.5)	3 (4.2)	1 (2.3)	
*KRAS p.G12A*	1 (0.9)	1 (1.4)	0	
*KRAS p.V14I*	1 (0.9)	0	1 (2.3)	
*HRAS p.G13R + KRAS p.Q61P*	1 (0.9)	0	1 (2.3)	

% valid percent; ^a^ Chi-square test; ^b^ Fisher’s exact test; Abbreviations: IHC—Immunohistochemical; n—number of patients; UBC—urothelial bladder cancer; UTUC—upper tract urothelial carcinoma.

**Table 3 biomedicines-12-02154-t003:** Univariate and multivariate Cox regression analysis of prognostic factors for PFS in UTUC patients stratified by luminal-basal subtypes.

Variables	Univariate Analysis			Multivariate Cox Regression Model		
	HR	CI 95%	*p*-Value	HR	CI 95%	*p*-Value
Gender (male ^a^ vs. female)	0.631	0.282–1.411	0.262			
Smoking status (never ^a^ vs. former/current)	1.017	0.437–2.369	0.969			
Hydronephrosis (no ^a^ vs. yes)	1.017	0.496–2.087	0.962			
Histological subtype (pure UC ^a^ vs. non-pure UC)	3.474	1.520–7.940	0.003	1.820	0.756–4.386	0.182
Tumor grade (low-grade ^a^ vs. high-grade)	2.863	0.383–21.399	0.305			
Lymphovascular invasion (no ^a^ vs. yes)	3.646	1.785–7.448	<0.001	3.265	1.378–7.734	0.007
Carcinoma in situ (no ^a^ vs. yes)	1.518	0.608–3.788	0.371			
AJCC staging system (0is−0a−I ^a^ vs. II−IV)	2.687	1.096–6.590	0.031	2.149	0.705–6.553	0.179
Lymph node involvement (no ^a^ vs. yes)	3.143	0.830–11.898	0.092			
Subgroups (only UTUC ^a^ vs. UTUC + UBC)	3.214	1.506–6.858	0.003	3.570	1.508–8.453	0.004
p53 (WT ^a^ vs. aberrant)	2.487	1.073–5.766	0.034	1.105	0.396–3.082	0.849
*FGFR3* (WT ^a^ vs. mutated)	0.521	0.241–1.126	0.097			
*TERTp* (WT ^a^ vs. mutated)	1.208	0.582–2.508	0.612			
*RAS* (WT ^a^ vs. mutated)	2.068	0.711–6.019	0.183			
*TP53* (WT ^a^ vs. mutated)	1.423	0.397–5.093	0.588			

Abbreviations: AJCC—American Joint Committee on Cancer; CI—confidence interval; HR—hazard ratio; UBC—urothelial bladder cancer; UC—urothelial carcinoma; UTUC—upper tract urothelial carcinoma; WT—wild-type; ^a^ Reference category.

**Table 4 biomedicines-12-02154-t004:** Univariate and multivariate Cox regression analysis of prognostic factors for OS in UTUC patients stratified by luminal-basal subtypes.

Variables	Univariate Analysis			Multivariate Cox Regression Model		
	HR	CI 95%	*p*-Value	HR	CI 95%	*p*-Value
Gender (male ^a^ vs. female)	0.552	0.226–1.348	0.192			
Smoking status (never ^a^ vs. former/current)	3.349	1.129–9.935	0.029	4.256	1.303–13.898	0.016
Hydronephrosis (no ^a^ vs. yes)	1.032	0.507–2.100	0.930			
Histological subtype (pure UC ^a^ vs. non-pure UC)	3.341	1.413–7.899	0.006	3.134	0.916–10.729	0.069
Tumor grade (low-grade ^a^ vs. high-grade)	1.041	0.302–3.593	0.949			
Lymphovascular invasion (no ^a^ vs. yes)	4.610	2.200–9.661	<0.001	13.510	3.837–47.570	<0.001
Carcinoma in situ (no ^a^ vs. yes)	1.394	0.568–3.426	0.469			
AJCC staging system (0is−0a−I ^a^ vs. II−IV)	1.360	0.643–2.878	0.421	1.937	0.649–5.780	0.236
Lymph node involvement (no ^a^ vs. yes)	3.650	1.094–12.178	0.035			
Subgroups (UTUC-only ^a^ vs. UTUC + UBC)	1.495	0.726–3.075	0.275	1.279	0.513–3.190	0.597
p53 (WT ^a^ vs. aberrant)	2.471	1.008–6.055	0.048	2.092	0.669–6.538	0.204
*FGFR3* (WT ^a^ vs. mutated)	0.823	0.402–1.687	0.595			
*TERTp* (WT ^a^ vs. mutated)	0.958	0.463–1.980	0.907			
***RAS*** (WT ^a^ vs. mutated)	1.926	0.659–5.632	0.231			
***TP53*** (WT ^a^ vs. mutated)	0.874	0.266–2.877	0.825			

Abbreviations: AJCC—American Joint Committee on Cancer; CI—confidence interval; HR—hazard ratio; UBC—urothelial bladder cancer; UC—urothelial carcinoma; UTUC—upper tract urothelial carcinoma; WT—wild-type ^a^ Reference category.

## Data Availability

The data presented in this study are available upon reasonable request from the corresponding author. The data are not publicly available due to patient privacy.

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
