# Peer review of "Biomarker Profiling of Upper Tract Urothelial Carcinoma Only and with Synchronous or Metachronous Bladder Cancer"

_biomedicines, 2024, doi:10.3390/biomedicines12092154_

Round 1

Reviewer 1 Report

Comments and Suggestions for Authors

the work is potentially interesting

changes are necessary to improve the quality of work

The 1st goal needs to be a little clearer

2. in the section material and methods, at the end of the paragraph on immunohistochemistry, add a reference as it was analyzed in other works: PMID: 30598340

3. In the discussion, p53 protein expression and gene regulation are mentioned, which do not correlate. It has been shown before that it is necessary to determine molecules on various cell neoplasms in tumors, which has been shown and suggested in the literature on the example of mediators of the immune system, but certainly cellular regulation from genes to membranes is complex and is studied as proteomics today, and it should be added and reference: PMID: 33131355

4. Discussion and conclusion should be corrected and changed

5. In the introduction, the chapter talks about genes, and in the conclusion, it says how immunohistology is good for analysis?

6. The introduction should be harmonized with the aim of the work and the conclusion, which is not the case in this case. One is the introduction and the conclusion is another.

Comments on the Quality of English Language

corrections

Author Response

Comment 1: The 1st goal needs to be a little clearer

Response 1: Thank you for your suggestion. Revised accordingly in the Section 1. Introduction (paragraph 4; lines 1-3).

Comment 2: In the section material and methods, at the end of the paragraph on immunohistochemistry, add a reference as it was analyzed in other works: PMID: 30598340

Response 2: According to the reviewer’s comment, we have included references in the manuscript in Section 2.4. Immunohistochemical Analysis, paragraph 3, lines 4 and 9 (references 17 and 18)

Comment 3: In the discussion, p53 protein expression and gene regulation are mentioned, which do not correlate. It has been shown before that it is necessary to determine molecules on various cell neoplasms in tumors, which has been shown and suggested in the literature on the example of mediators of the immune system, but certainly cellular regulation from genes to membranes is complex and is studied as proteomics today, and it should be added and reference: PMID: 33131355

Response 3: We agree with the reviewer’s comment. Discussion has been revised in Section 4. Discussion (paragraph 9; lines 8-11) and reference was added in Section. References, reference 39).

Comment 4: Discussion and conclusion should be corrected and changed

Response 4: Thank you very much for your suggestion. This has been addressed.

Comment 5: In the introduction, the chapter talks about genes, and in the conclusion, it says how immunohistology is good for analysis?

Response 5: Thank you for your relevant remark. Our manuscript has been revised and improved in the Section 1. Introduction (paragraph 2-4) and Section 5. Conclusion (paragraph 2).

Comment 6: The introduction should be harmonized with the aim of the work and the conclusion, which is not the case in this case. One is the introduction and the conclusion is another.

Response 6: Thank you very much for your remark. Introduction and Conclusion have been improved. Please take special consideration to Section 1. Introduction (paragraph 2-4) and to Section 5. Conclusion (paragraph 2).

Reviewer 2 Report

Comments and Suggestions for Authors

The manuscript titled “Biomarker Profiling of Upper Tract Urothelial Carcinoma Only and with Synchronous or Metachronous Bladder Cancer” by Meireles, investigated the molecular differences between upper tract urothelial carcinoma alone and UTUC with synchronous or metachronous urothelial bladder cancer. A cohort was analyzed based on immunohistochemical (IHC) expression of CK5/6, CK20, GATA3, and p53, and genetic characterization including TETRTp. FGFR3, TP53, and RAS. Results showed no significant differences in luminal-basal subtypes or genetic profiles between UTUC-only and UTUC+UBC. This is a novel aspect in a field of intense research. Still some issues need to be clarified, as listed below, before the manuscript can be accepted for publication in Biomedicines.

11.      Please provide a more detailed explanation of selection criteria for UTUC and UBC cases. Do the samples represent the general characteristics of UTUC and UBC patients? Was the diversity of the sample and the clinical background taken into consideration?

22.      The study finds that FGFR3 mutations are associated with better clinical outcomes. Could you provide more detailed mechanistic explanations regarding how FGFR3 mutations impact the prognosis of UTUC?

33.      The study mentions that IHC features are strongly correlated with luminal-basal subtypes. What specific experimental design or data collection methods do you suggest to further validate the role of IHC in the stratification of UTUC?

44.      Although the results indicate that UTUC cases alone and those coexisting with UBC are similar in phenotypic and genomic features, could you provide more detailed molecular-level analyses to support these similarities?

Author Response

Comment 1: Please provide a more detailed explanation of selection criteria for UTUC and UBC cases. Do the samples represent the general characteristics of UTUC and UBC patients? Was the diversity of the sample and the clinical background taken into consideration?

Response 1: Thank you for your relevant comments. The authors reported, so far, the most extensive single-center retrospective data in Portugal. The sample accurately represents the clinical reality and characteristics of the national target population. We collected a dataset of tissue samples from a cohort of patients diagnosed with UTUC in our institution between January 2009 and December 2019 and submitted to radical nephroureterectomy resection or kidney-sparing approach. “Synchronous” UBC tumors refer to cases in which the second primary cancer was diagnosed within six months of the primary UTUC, and “Metachronous” UBC refers to instances in which the diagnosis was more than six months after the primary UTUC (please take special attention to Section 2.1 Study design, paragraph 1).

Comment 2: The study finds that FGFR3 mutations are associated with better clinical outcomes. Could you provide more detailed mechanistic explanations regarding how FGFR3 mutations impact the prognosis of UTUC?

Response 2: Thank you for your remark. According to the reported data, FGFR3 mutations were associated with non-invasive, low-stage tumors and a milder disease course in UTUC, and invasive UTUCs with FGFR3 mutations have a more favorable prognosis. The authors have already mentioned references. Please consider references 29, 30, and 33 cited in Section 4. Discussion, paragraph 6, line 9.

Comment 3: The study mentions that IHC features are strongly correlated with luminal-basal subtypes. What specific experimental design or data collection methods do you suggest to further validate the role of IHC in the stratification of UTUC?

Response 3: Thank you for your comment. It would be interesting to perform a more extensive molecular analysis and large-scale DNA and RNA sequencing techniques with high-throughput methodology (next-generation sequencing). However, one of the aims of this study was to validate luminal-basal stratification in UTUC through less complex and cost-effective methods with applicability in routine clinical practice. Please see specified in Section 4. Discussion (paragraph 4).

Comment 4:  Although the results indicate that UTUC cases alone and those coexisting with UBC are similar in phenotypic and genomic features, could you provide more detailed molecular-level analyses to support these similarities?

Response 4: Thank you very much for this pertinent question. This has already been addressed by the authors, considering that further research focused on classifying distinct immune subtypes and investigating the relationships of tumor-infiltrating immune cells and molecular signatures on the tumor microenvironment of UTUC might be crucial for a deeper understanding of tumor biology. Please see specified in Section 4. Discussion (paragraph 19; lines 1-3)

Reviewer 3 Report

Comments and Suggestions for Authors

General: this study focused on the molecular profiling and biomarker exploratory of UTUC-UBC, a rare type of malignancy with no genomic data has been revealed previously. This study is of certain clinical value. But some major issues remained to be addressed further.

Major:

1.     This analysis compared two subgroups including UTUC-only and UTUC+UBC. In UTUC+UBC, the authors included UTUC with synchronous or metachronous UBC. What is the molecular and survival differences between UTUC with synchronous and with metachronous UBC? Therefore, a further subgroup analysis in the UTUC+UBC subtype is needed.

2.     The author also mentioned that “metachronous bladder tumors appear to uphold the molecular features of the initial UTUC” (line 59), are there any relevant evidence provided in this article?

3.     In the genetic profiling data, the specific types of mutation were provided. Is there more information about the copy number variation of these genes? And how many genes were detected in total in the targeted sequencing, only four genes? Compared with another article published in 2021 (PMID:32861617), a targeted sequencing containing a larger scale of genetic information should be added. As four genes provided little information, and all of them showed no association with PFS or OS in the supplemental file. Therefore, to obtain more genetic information of this rare malignancy, at least a small panel of 50-100 tumor driver gene, is recommended. Regarding to the visualization of mutated genes, an oncoplot containing all the mutation information is more standard and therefore recommended.

Minor:

1.     43 participants with UTUC+UBC were recruited. However, in Table 2, the total number in some markers and genes were inconsistent (e.g. n=42 in TP53, n=37 in luminal-basal phenotype). Therefore, the information of undetected sample should be provided.

Author Response

Major:

Comment 1: This analysis compared two subgroups including UTUC-only and UTUC+UBC. In UTUC+UBC, the authors included UTUC with synchronous or metachronous UBC. What is the molecular and survival differences between UTUC with synchronous and with metachronous UBC? Therefore, a further subgroup analysis in the UTUC+UBC subtype is needed.

Response 1: We recognize the value of the reviewer’s suggestion. However, the diagnostic criteria for synchronous and metachronous tumors can be misleading, sometimes overlapping the cutoff (3, 6 months). Besides, a crucial unanswered question is whether temporally different UTUC and UBC tumors, developing in the same patient with no previous UBC history, represent a genomic-related intravesical recurrence or a divergent primary tumor (Wang Y et al. 2013, PMCID: PMC3765602). As expected, we found a lower representativity of the synchronous subgroup (n=14), which would make the analysis statistically underpowered. These limitations lead us to include both subgroups, increasing our data's statistical power and clinical representativeness.

Comment 2: The author also mentioned that “metachronous bladder tumors appear to uphold the molecular features of the initial UTUC” (line 59), are there any relevant evidence provided in this article?

Response 2: Thank you for your relevant question. The authors have recently published a molecular analysis of a non-muscle invasive bladder cancer cohort showing an equivalent frequency of TERTp and FGFR3 mutations in UBC samples (detailed in Section 4. Discussion, paragraph 10, lines 1-4). Published data on this subject usually comprised non-muscle invasive and muscle-invasive tumors, making this comparison challenging.

Comment 3: In the genetic profiling data, the specific types of mutation were provided. Is there more information about the copy number variation of these genes? And how many genes were detected in total in the targeted sequencing, only four genes? Compared with another article published in 2021 (PMID:32861617), a targeted sequencing containing a larger scale of genetic information should be added. As four genes provided little information, all of them showed no association with PFS or OS in the supplemental file. Therefore, to obtain more genetic information of this rare malignancy, at least a small panel of 50-100 tumor driver gene, is recommended. Regarding to the visualization of mutated genes, an oncoplot containing all the mutation information is more standard and therefore recommended. 

Response 3: The reviewer's comments are appropriate, and we acknowledge the suggestions. Many studies on genomic profiling of UTUC have been reported (references 10-13). However, a highly complex and costly advanced molecular technology limits a comprehensive molecular analysis's applicability in clinical practice settings. Sjodahl et al. concluded that a taxonomic classification system incorporating tumor cell phenotype using immunohistochemistry and gene expression analysis would be more realistic to bridge the gap between molecular-based classification and traditional morphology/immunohistochemistry-based pathologic classification in urothelial carcinoma (reference 16, PMID: 22553347). IHC has been explored as a readily accessible, relatively inexpensive, standardized tool for molecular subtyping in clinical practice.
Furthermore, IHC has the advantage of disclosing the location of single markers, thus allowing to distinguish between signals from non-tumor cells (i.e., stromal cells, immune cells) and those from cancer cells, which may strictly affect the reliability of molecular subtyping based on gene-expression profiling, especially in the setting of highly infiltrated tumors (reference 20, PMID: 35887192). We also realize that our study's genetic characterization of UTUC presents the most relevant mutations reported in urothelial carcinoma and 76.3% of all UTUC samples were mutated for at least one of the studied genes. Besides, our cohort represents a single-center retrospective study, and tumors are frequently heterogeneous, which might explain the results regarding FGFR3, RAS, and TERTp mutations in PFS and OS in this analysis.

Please see the specified aims of our study in Section 1. Introduction (paragraph 4; lines 1-3) and reference mentioned in SectionReferences. 16 and 20. Take special attention to oncoplot added with all the mutation profile data in Figure 2.

Minor:

Comment 1: 43 participants with UTUC+UBC were recruited. However, in Table 2, the total number in some markers and genes were inconsistent (e.g. n=42 in TP53, n=37 in luminal-basal phenotype). Therefore, the information of undetected sample should be provided.

Response 1: Thank you for pointing this out. Missing data can be due to a lack of available information or data technically infeasible regarding IHC markers or genes analysis (Please see specified in Section 2.6. Outcomes and Statistical Analysis, paragraph 5, lines 1-2). The valid percentage was included in all tables, considering non-missing data.

Round 2

Reviewer 1 Report

Comments and Suggestions for Authors

the authors made minimal corrections and did not respond to all suggestions

need to add appropriated citation

Comments on the Quality of English Language

correction is need

Author Response

Reviewer 1 (Round 2)

Comments and Suggestions for Authors: The authors made minimal corrections and did not respond to all suggestions. Need to add appropriated citation

Response: Thank you very much for your comments. Here, below, we have addressed point-by-point all your previous suggestions and concerns.

Comment 1: The 1st goal needs to be a little clearer

Thank you for your suggestion. Revised accordingly in the Section 1. Introduction (paragraph 4; lines 1-3).

“This study intends to identify biological differences among patients with UTUC-only and UTUC with synchronous or metachronous bladder cancer (UTUC-UBC) based on IHC and gene expression analysis. We also aim to understand the putative effect of synchronous or metachronous bladder cancer on the outcome of UTUC patients after adjusting for luminal-basal phenotype with an immunohistochemistry-based protocol.” 

Comment 2: In the section material and methods, at the end of the paragraph on immunohistochemistry, add a reference as it was analyzed in other works: PMID: 30598340

According to the reviewer’s comment, we have included references (references 17 and 18) in the manuscript in Section 2.4.Immunohistochemical Analysis (paragraph 3, lines 4 and 9)

The reference suggested by the reviewer (PMID: 30598340) specifies a IHC score of HER2 in breast cancer. The authors cited new references about IHC scores in urothelial carcinoma (UC), once IHC score for breast cancer does not apply for UC.

“The expression for CK5/6, CK20, and GATA3 was evaluated semi-quantitatively according to the staining intensity (absent = 0, faint = 1, moderate = 2, or strong = 3) and proportion of positive-stained tumor cells (scored as <5% = 0; 5–25% = 1; 25–50% = 2; 50–75% = 3; and >75% = 4) [17].”

“Cases stained for p53 were classified as follows: wild type (1–49 % nuclear expression) or aberrant (null-phenotype: 0% nuclear expression; 50-99% nuclear expression; or diffuse overexpression: 100% nuclear expression) [18].

Comment 3: In the discussion, p53 protein expression and gene regulation are mentioned, which do not correlate. It has been shown before that it is necessary to determine molecules on various cell neoplasms in tumors, which has been shown and suggested in the literature on the example of mediators of the immune system, but certainly cellular regulation from genes to membranes is complex and is studied as proteomics today, and it should be added and reference: PMID: 33131355

We agree with the reviewer’s comment. Discussion has been revised in Section 4. Discussion (paragraph 9; lines 8-11) and reference suggested by the reviewer was added in Section. References (reference 39).

“Otherwise, we must consider the high complexity of cellular processes regulation, and molecular and tumor microenvironment interactions [39]. Further proteome-wide comprehensive analysis of p53 IHC pattern would be necessary to better identify and acknowledge any such TP53 dysfunctions.”

The authors also have highlighted in the manuscript the relevance of further research focused on urothelial carcinoma tumor microenvironment.

“Likewise, further research focused on UC’s tumor microenvironment will help us better understand tumor biology and how the immune landscape reflects on clinical outcomes and immunotherapy responses. “

Due to the strong heterogeneity of urothelial carcinoma, we could have distinct outcomes and immunotherapy responses. Most of subtyping methods, proposed in published data, are based on a limited number of biomarkers, and none of them is developed on the basis of cell states. We believe that sequencing at the transcriptomic and genomic level does not provide any information about protein–protein interactions and protein localization, which are critical to understand numerous signaling pathways.  A comprehensive analysis of multiomics data from UTUC patients could help us to better understand disease-specific regulatory mechanisms. This topic was beyond the scope of this manuscript.

Comment 4: Discussion and conclusion should be corrected and changed

Thank you very much for your valuable suggestion. This has been addressed accordingly your additional comments.

“Otherwise, we must consider the high complexity of cellular processes regulation, and molecular and tumor microenvironment interactions [39]. Further proteome-wide comprehensive analysis of p53 IHC pattern would be necessary to better identify and acknowledge any such TP53 dysfunctions.”

“Besides, immunohistochemistry has the unique advantage of disclosing the precise morphological distribution of biomarkers in cell/tissue component [20].”

“Nonetheless, an immunohistochemistry-based method for a critical assessment of luminal-basal stratification in UTUC still needs to be approved for routine clinical practice.”

Comment 5: In the introduction, the chapter talks about genes, and in the conclusion, it says how immunohistology is good for analysis?

Thank you for your relevant remark. Our manuscript has been revised and improved in the Section 1. Introduction (paragraph 2-4) and Section 5. Conclusion (paragraph 2).

As cited below, the authors debate the relevance of a comprehensive classification including IHC and genetic expression as being more effective in subtyping UTUC. Thus, our study has included both analyses to answer our main aims (biological comparison between UTUC vs UTUC+UBC and the effect of UBC in outcome of UTUC patients). However, a secondary aim was to stratify these patients taking into account luminal-basal classification and validate IHC as an easier tool in clinical practice to subtype these patients.

“There is no substantial biomarker profiling data in primary UTUC with synchronous or metachronous UBC. Indeed, a stratification system in UTUC patients incorporating immunohistochemical (IHC) phenotype and gene expression analysis might be more reasonable to overcome the gap between molecular-based classification and the classic pathologic/immunohistochemistry-based classification [16]. However, the application of immunohistochemistry for a basic assessment of luminal-basal stratification in UTUC patients is pending for use in clinical practice.” 

“Our study reveals a similar luminal-basal phenotype and genetic features among patients with UTUC with synchronous or metachronous UBC and UTUC without UBC. An immunohistochemistry-based protocol was recognized as an accessible and standardized method for luminal-basal classification in UTUC patients.Furthermore, in the era of molecular subtypes, the association of bladder cancer does not seem to convey a worse outcome for UTUC patients, even when adjusted for molecular and luminal-basal subtypes.”

Comment 6: The introduction should be harmonized with the aim of the work and the conclusion, which is not the case in this case. One is the introduction and the conclusion is another.

Thank you very much for your suggestion. Introduction and Conclusion have been improved. Please take special consideration to Section 1. Introduction (paragraph 2-4) and to Section 5. Conclusion (paragraph 2).

Accordingly, to the reviewer’s comment, the manuscript has been revised in Introduction as cited below. The authors believe that the conclusions are now aligned with the aims mentioned in introduction.

“Indeed, a stratification system in UTUC patients incorporating immunohistochemical (IHC) phenotype and gene expression analysis might be more reasonable to overcome the gap between molecular-based classification and the classic pathologic/immunohistochemistry-based classification [16]. However, the application of immunohistochemistry for a basic assessment of luminal-basal stratification in UTUC patients is pending for use in clinical practice."

“Our study reveals a similar luminal-basal phenotype and genetic features among patients with UTUC with synchronous or metachronous UBC and UTUC without UBC. An immunohistochemistry-based protocol was recognized as an accessible and standardized method for luminal-basal classification in UTUC patients. Furthermore, in the era of molecular subtypes, the association of bladder cancer does not seem to convey a worse outcome for UTUC patients, even when adjusted for molecular and luminal-basal subtypes.”

Comments on the Quality of English Language: correction is need

Thank you for your recommendation. Since authors are not native-born English speakers, an independent review by a native-born English writer was performed previously to submission and no major issues were raised. This was also the comment of the other two reviewers.  

Additional References:

  • Sjodahl, G.; Lauss, M.; Lovgren, K.; Chebil, G.; Gudjonsson, S.; Veerla, S.; Patschan, O.; Aine, M.; Ferno, M.; Ringnér, M; et al. A molecular taxonomy for urothelial carcinoma. Clin Cancer Res. 2012, 18, 3377–86. DOI: 10.1158/1078-0432.
  • Liu, H.; Shi, J.; Wilkerson, M.L.; Lin, F. Immunohistochemical evaluation of GATA3 expression in tumors and normal tissues: a useful immunomarker for breast and urothelial carcinomas. Am J Clin Pathol. 2012, 138, 57-64. DOI: 10.1309/AJCP5UAFMSA9ZQBZ.
  • Hodgson, A.; Xu, B.; Downes, M.R. p53 immunohistochemistry in high-grade urothelial carcinoma of the bladder is prognostically significant. Histopathology. 2017, 71, 296-304. DOI: 10.1111/his.13225.
  • Jurisic V. Multiomic analysis of cytokines in immuno-oncology. Expert Rev Proteomics. 2020, 17, 663-674. DOI: 10.1080/14789450.2020.1845654.
  • Sanguedolce, F.; Zanelli, M.; Palicelli, A.; Ascani, S.; Zizzo, M.; Cocco, G.; Björnebo, L.; Lantz, A.; Landriscina, M.; Conteduca, V.; et al. Are We Ready to Implement Molecular Subtyping of Bladder Cancer in Clinical Practice? Part 2: Subtypes and Divergent Differentiation. Int J Mol Sci. 2022, 23, 7844. DOI: 10.3390/ijms23147844.

Reviewer 3 Report

Comments and Suggestions for Authors

The questions have been well-addressed. A standard form to visualize genetic mutation has been added by the authors. Therefore the quality of this article has been improved.

Author Response

Reviewer 3 (Round 2)

Comments and Suggestions for Authors: The questions have been well-addressed. A standard form to visualize genetic mutation has been added by the authors. Therefore the quality of this article has been improved.

We are sincerely thankful for the Reviewer’s overall comment. We are truly pleased with his constructive feedback to improve our manuscript.

Round 3

Reviewer 1 Report

Comments and Suggestions for Authors

corrected acording sugestions

Comments on the Quality of English Language

corrected

Author Response

Thank you very much for handling our paper with your best care and getting it reviewed for publication in Biomedicines. It were your insightful comments that led us to improve our current version.